# Principal Component Analysis to Assess the Changes of Yield and Quality in *Pinellia ternata* at Different Stages after Brassinolide Treatments

**DOI:** 10.3390/ijms232315375

**Published:** 2022-12-06

**Authors:** Chenchen Guo, Yanfen Zhang, Dengyun Wu, Mengyue Wang, Yu Du, Jianzhou Chu, Xiaoqin Yao

**Affiliations:** 1School of Life Sciences, Hebei University, Baoding 071002, China; 2Institute of Life Sciences and Green Development, Hebei University, Baoding 071002, China; 3Key Laboratory of Microbial Diversity Research and Application of Hebei Province, Baoding 071002, China; 4Technology Transfer Center, Hebei University, Baoding 071002, China

**Keywords:** brassinolide, growth stages, *Pinellia ternata*, principal component analysis

## Abstract

Brassinolide (BR) is the “sixth class” plant hormone, which plays an important role in various physiological and biochemical processes of plants. The wide variety of functions of *Pinellia ternata* means that there is huge demand for it and thus it is in short supply. This paper mainly assessed the changes of yield and quality in *P. ternata* at different stages after BR treatments by principal component analysis, in order to improve the yield and quality of *P. ternata* and at the same time determine the best harvest time. The results showed that the tuber yield of *P. ternata* was significantly increased by BR treatments at different stages (except for the 15th day). After the 15th, 45th, 60th, 75th, 90th, and 105th day of treatments, the tuber yield of *P. ternata* reached peak values at 0.10 (0.65 g), 0.50 (1.97 g), 0.50 (1.98 g), 1.00 (2.37 g), 1.00 (2.84 g), and 2.00 mg/L (3.76 g) BR treatment, respectively. The optimal harvest time was the 75th day after 0.10, 0.50, and 1.00 mg/L BR treatments, which not only significantly improved the yield of *P. ternata*, but also retained high level of total alkaloids in the tubers (20.89, 5.37, and 13.44%) and bulbils (9.74, 20.42, and 13.62%), high total flavone content in the tubers (17.66, 16.26, and 12.74%) and bulbils (52.63, 12.79, and 38.69%), and high β-sitosterol content in the tubers (25.26, 16.65, and 0.62%) of *P. ternata*, compared with the control, respectively.

## 1. Introduction

*Pinellia ternata* is a traditional herbal medicine of China and Japan, which can be dated back to 2000 years ago in China. The tubers and bulbils of *P*. *ternata* contain alkaloids, β-Sitosterol, flavonoids, lectins, volatile oils, fatty acids, etc., among which alkaloids are most important active components. Pharmacological activity has n that *P*. *ternata* alkaloids inhibited the growth of hepatocarcinoma cell, ascitic mice, HeLa cells, gastric cancer cells, chronic myeloid leukemia cells, and other cancer cells [1]. β-sitosterol is a promising, safe, and low toxic anticancer drug, which was isolated from *P*. *ternata*. Flavonoids isolated from *P*. *ternata* have inhibitory activity on aldose reductase, which could be used to treat diabetes mellitus. Moreover, the protein and soluble sugar isolated from *P*. *ternata* also have biological activities, such as anti-early pregnancy effect and antiemetic [1,2]. At present, *P*. *ternata* is used as a raw material to produce medicines, such as “Banxia Houpu Decoction” “Sho-seiryu-to” “Kakkon-to” and “Choto-san”, these have been used clinically to treat a variety of diseases [3,4]. Moreover, the toxicity of *P*. *ternata* extracts to multiple pests makes it possible for biological pesticides [5,6]. Due to the wide variety of functions of *P*. *ternata*, there is a huge demand for it, meaning it is in short supply. However, there are few studies on improving the yield and quality of *P*. *ternata*. For example, light intensities and temperatures impacted the growth and tube yield of *P*. *ternata* [7,8]. *P*. *ternata* mainly reproduces asexually. Moreover, there are few studies on the bulbil of *P*. *ternata*, which is a crucial propagative material of *P*. *ternata*. The wild resource of *P*. *ternata* is rare and the technology is also deficient in the artificial cultivation of *P*. *ternata*. Therefore, it is necessary to improve the yield and quality of *P*. *ternata* at the same time.

Brassinolide (BR) is a plant hormone which plays an important role in various physiological and biochemical processes of plants, such as cell division and elongation, regulation of gene expression, vascular differentiation, reproductive development, and photosynthesis [9,10,11]. Several studies have shown that BR increases plant growth under various environmental stresses, such as salt stress [12], organic pollutants [13], water stress [14], and weak light stress [15]. *P*. *ternata* is susceptible to wither caused by environmental stress, resulting in the decline of the yield and quality of *P*. *ternata*. The most classical breeding and genome editing are often used to achieve plant resistance to abiotic stress. However, their development is limited by time-consuming or moral considerations [16]. Plant growth regulators have been explored to improve the growth of various crops [17]. BRs can inhibit the growth of several human cancer cell lines (breast and prostate) at micromolar concentrations without affecting the growth of normal cells [18]. BR is non-toxic and prevents the growth of cancer cells, and they can be used in field cultivation without causing environmental pollution. Understanding the effect of different BR concentrations on the yield and quality of *P*. *ternata* is very important in field cultivation.

This paper mainly assessed the changes of yield and quality in *P*. *ternata* at different stages after BR treatments by principal component analysis, in order to improve yield and quality of *P*. *ternata* at the same time and to determine the best harvest time. We chose the optimal model for harvesting time after different concentrations of BR treatment, which would improve yield in the artificial cultivation of *P*. *ternata*.

## 2. Results

### 2.1. Effects of BR Treatment on Yield of P. ternata at Different Stages

BR application had a noticeable effect on the tuber yield of *P*. *ternata* at different stages. As shown in Figure 1, the tuber yield of *P*. *ternata* was decreased by 14.90, 15.16, and 28.50% on the 15th day after 0.05, 0.50, and 1.00 mg/L BR treatments, compared with the control, respectively. The 0.05 mg/L BR treatment raised the tuber yield of *P*. *ternata* by 19.80, 23.99, and 48.00% on the 45th, 60th, and 75th day after treatment, compared with the control, respectively. After the 90th and 105th day of treatments, 0.10 mg/L BR treatment enhanced the tuber yield of *P*. *ternata* by 78.99 and 52.16%, compared with the control, respectively. The tuber yield of *P*. *ternata* was enhanced by 59.38, 42.80, 27.51, and 16.77% on the 45th, 60th, 90th, and 105th day after 0.50 mg/L BR treatment, compared with the control, respectively. After the 60th, 75th, and 90th day of treatments, 1.00 mg/L BR treatment raised the tuber yield of *P*. *ternata* by 29.03, 47.77, and 80.82%, compared with the control, respectively. The 2.00 mg/L BR treatments increased the tuber yield of *P*. *ternata* by 32.51, 60.53, and 93.90% on the 75th, 90th, and 105th day after treatment compared with the control, respectively. After the 15th, 45th, 60th, 75th, 90th, and 105th of BR treatments, the tuber yield of *P*. *ternata* reached the peak value at 0.10, 0.50, 0.50, 1.00, 1.00, and 2.00 mg/L BR, respectively. 

### 2.2. Extraction of Principal Components

BR treatments had significant effects on the quality of tubers and bulbils of *P*. *ternata* at different stages (Appendix A). In order to completely analyze the effect of BR treatments on tubers and bulbils of *P*. *ternata* at different stages, a comprehensive index for evaluating total flavone, total alkaloids, β-sitosterol, soluble protein, soluble sugar, free amino acid, ascorbic acid content, and DPPH radical scavenging activity in the tubers and bulbils of *P*. *ternata* was established by PCA. The first four principal components account for 75.67% of the total variance in the dataset. As seen in Figure 2, PC1 and PC2 were positively correlated with soluble protein, free amino acids, and soluble sugar in the tubers and bulbils, and ascorbic acid in the tubers of *P*. *ternata*, which explained 23.97% and 18.47% of the total variation, respectively. PC1 and PC2 were related to the soluble protein, free amino acids, soluble sugar, and other nutrient components in tuber and bulbil of *P*. *ternata*. PC3 and PC4 were positively correlated with total flavone, total alkaloids, and β-sitosterol in tubers and bulbils, and ascorbic acid with the bulbils of *P*. *ternata*, which explained 16.98% and 16.25% of the total variation, respectively (Figure 3). PC3 and PC4 were related to the active component in the tubers and bulbils of *P*. *ternata*.

### 2.3. PC1-PC2 Loading and Scores for Different Stage after BR Treatments

Figure 2B shows the scores of PC1 and PC2 at different stages after BR treatments. PC1 scores on the 105th day after BR treatments were higher than those at the other stages. After the 45th, 60th, 90th, and 105th day of treatments, BR treatments (except for E4, 1 mg/L BR on the 90th day) enhanced the PC1 score. The PC1 score was only increased by low-concentration (A1, 0.05 mg/L; A2, 0.10 mg/L) BR on the 15th day and low-concentration (D1, 0.05 mg/L) and high-concentration (D4, 1.00 mg/L; D5, 2.00 mg/L) BR on the 75th day after treatments. PC2 scores on the 60th day after BR treatments were higher than those at the other stages. After the 15th, 45th, 60th, 75th, and 90th days of treatments, low-concentration BR (0.05 mg/L or 0.10 mg/L) improved the PC2 score compared with the control. After the 105th day of treatments, the PC2 score was increased by BR treatments. 

### 2.4. PC3-PC4 Loading and Score for Different Stage after BR Treatments

The changes in PC3-PC4 score after BR treatments were different at different stages (Figure 3B). After the 15th and 75th day of treatments, the PC3 and PC4 scores were enhanced by BR treatments, and the PC3-PC4 score on the 75th day was higher than that at the other stages. However, the PC3 score did not increase on the 45th, 60th, 90th, and 105th day after BR treatments (except for 0.10 mg/L BR on the 60th day) when compared with the control. After the 45th day of treatments, only low-concentration BR (B1, 0.05 mg/L; B2, 0.10 mg/L; B3, 0.50 mg/L) increased the PC4 score compared with the control. After the 90th day of treatments, the PC4 score was enhanced by BR treatments (except for E4, 1.00 mg/L). Moreover, 0.50 mg/L BR on the 60th day (C3) and 2.00 mg/L BR on the 105th day (F5) increased the PC4 score compared with the control.

### 2.5. Effect of BR Treatments on PAL and GS Activity of P. ternata at Different Stages

Figure 4A showed that the PAL activity in the tubers of *P*. *ternata* was increased by 22.66, 14.30, and 73.76% on the 45th, 60th, and 75th day after 0.05 mg/L BR treatment, compared with the control, respectively. After the 15th, 45th, and 60th day of treatments, the PAL activity in the tubers of *P*. *ternata* was enhanced by 18.20, 33.02, 32.48% in 0.10 mg/L BR treatment, and was raised by 29.48, 11.76, and 60.04% in 0.50 mg/L BR treatment, compared with the control, respectively. The 1 mg/L BR treatments increased PAL activity in the tubers of *P*. *ternata* by 21.50, 20.84, 28.99, 14.49, 38.24, and 39.90% during growth stages, compared with the control, respectively. After the 15th, 60th, 75th, 90th, and 105th day of treatments, 2.00 mg/L BR treatment enhanced PAL activity in tubers of *P*. *ternata* by 25.72, 37.90, 50.13, 29.68, and 65.37%, compared with the control, respectively. After the 15th and 75th day of treatments, PAL activity in the bulbils of *P*. *ternata* was increased by 5.77 and 65.52% in 0.05 mg/L BR treatment and was raised by 29.64 and 23.89 in 0.10 mg/L BR treatment, compared with the control, respectively (Figure 4B). The 0.50 mg/L BR treatment increased PAL activity (104.83%) in the bulbils of *P*. *ternata* on the 90th day after treatment. PAL activity in the bulbils of *P*. *ternata* was raised by 16.32, 33.27, 25.54, and 5.26% on the 15th, 75th, 90th, and 105th day after 1.00 mg/L BR treatment, and was enhanced by 5.96 and 13.54% on the 60th and 90th day after 2.00 mg/L BR treatment, compared with the control, respectively.

GS activity in the tubers of *P*. *ternata* was enhanced by 7.18, 5.35, and 21.33% on the 15th, 60th, and 105th day after 0.10 mg/L BR treatments, and was increased by 13.42% on the 105th day after 0.50 mg/L BR treatments, compared with the control, respectively (Figure 5A). After the 15th and 90th day of treatments, 1 mg/L BR treatment raised GS activity in the tubers of *P*. *ternata* by 12.72 and 14.87%, compared with the control, respectively. The 2.00 mg/L BR treatment enhanced GS activity (5.38%) in the tubers of *P*. *ternata* on the 60th day after treatment compared with the control. GS activity in the bulbils of *P*. *ternata* was increased by 10.38% on the 45th day after 0.05 mg/L BR treatment and was raised by 19.90 and 14.45% on the 15th and 60th day after 0.10 mg/L BR treatment, compared with the control, respectively (Figure 5B). After the 15th day of treatments, GS activity in the bulbils of *P*. *ternata* was increased by 1.00 mg/L (9.50%) BR treatment compared with the control.

## 3. Discussion

Under abiotic stress, experiments have demonstrated that BR often improves plant yield and quality. However, most of these studies were on crop plants grown in abiotic stress conditions on the effects of nutritional quality. There were no studies on the effects of BR treatment on the yield and active ingredients of medicinal plants during growth processes. It is clearly very important to know the optimal BR levels and harvest stages for growth and active ingredients accumulation of plant. In this study, on the 75th day after BR treatments, the optimal BR concentrations were 0.10, 0.50, and 1.00 mg/L, which not only significantly improved the yield of *P*. *ternata*, but also retained a high level of active components. We chose the optimal model for harvesting time after different concentrations of BR treatment.

We found that the application of BR had a significant effect on the tuber yield of *P*. *ternata* at different stages (Figure 1). Previous studies showed that BR treatment increased plant yields of maize [19], potato [20], pepper, wheat, rice, groundnut, mustard, and cotton [21]. Serna et al. [22] also reported that spraying BR analogs increased the yield of field-grown pepper, and increased yield caused by an increase in pepper/plant number. We observed that the tuber yield was significantly increased by BR treatments at different stages (except for the 15th day). This indicated that BR treatments could improve early growth of *P*. *ternata* by rapidly consuming nutrients in the tuber, which resulted a decrease in tuber yield on the 15th day after BR treatments. Others have reported that BR treatment increased fruit weight in fruits such as yellow passion fruit [23] and lychee [24]. The increased tuber yield might be related to the improvement of leaf carbon assimilation rate by BR treatment [25]. 

PCA is an effective mathematical method to reduce the dimension of multivariate data while retaining most of the variance. Liu et al. [26] reported that the effect of ellagic acid on the storage of kumquats was evaluated by PCA. In this study, the most important PC1 and PC2 were extracted, which contained nutrient components such as soluble protein, free amino acids, and soluble sugar in the tubers and bulbils of *P*. *ternata*. Nutrient components are crucial for plant growth and metabolism. Compared with other concentrations, low-concentration (0.05 and 0.10 mg/L) and high-concentration (1.00 and 2.00 mg/L) BR treatments showed remarkable improvement in nutrient components in the tubers and bulbils of *P*. *ternata*. In other studies, the application of 0.48 mg/L BR increased soluble sugar content in wheat and mustard [27]. The research performed by Yusuf et al. [28] showed that BR treatment increased soluble sugar content in two wheat cultivars. These results were similar to ours in that soluble sugar content in the tubers and bulbils of *P*. *ternata* was enhanced on the 15th, 60th, and 105th day after 0.50 mg/L BR treatment. The content of soluble sugar in tubers and bulbils of *P*. *ternata* reached a peak value on the 60th day. The reason for this might be that the new tiller needed tubers to supply nutrients on the 75th day after BR treatments.

We found that BR treatments improved the soluble protein and free amino acid content in the tubers and bulbils of *P*. *ternata* during growth processes. This was similar to results obtained in previous studies on rice [29,30] and tomatoes [31]. After the 105th day of treatments, BR showed a higher soluble protein and free amino acid content compared with other stages. The reason for this might be that on the 105th day after treatments, the new tiller plants began to accumulate material, which increased the soluble protein and free amino acid content. However, the nutrient supply of new tillers led to the decrease of soluble sugar content. BR treatment elevated soluble protein and free amino acid content levels, which might be the result of BR analogues’ effects on transcription and/or translation, thus altering the pattern of soluble proteins and enzymes [32]. 

Ascorbic acid is an important non-enzymatic antioxidant in plants, which is mainly involved in the scavenging of reactive oxygen species. DPPH radical scavenging activity reflects plants’ antioxidant capacity. The present study revealed the promoting effect of BR treatments on ascorbic acid content and DPPH radical scavenging activity in the tubers and bulbils of *P*. *ternata*. This result was supported by the findings of Mussig et al. [33], in which the expression of gen for monodehydroascorbate reductase was regulated by BR. Similar results also found that 0.048 mg/L epibrassinolide treatment significantly raised ascorbic acid content (47.26%) and DPPH radical scavenging activity (146.17%) in radish seedlings when compared with control [34]. The increase in DPPH radical scavenging activity might be related to a high level of flavonoids after BR treatments [13]. 

Total alkaloids, β-sitosterol, and total flavone have been considered as the active ingredients in the tubers and bulbils of *P*. *ternata* [1]. We found that BR treatments significantly increased the content of active ingredients in the tubers and bulbils of *P*. *ternata*, and the content of active ingredients in the tubers and bulbils of *P*. *ternata* on the 75th day after BR treatments was higher than that of the other stages (Figure 3). Similar studies had shown that epibrassinolide treatments significantly increased glycine betaine content in tomato leaves [35]. The initial steps of alkaloid synthesis come from amino acids, and their biosynthesis includes multistep reactions [36]. In our study, BR treatments increased the free amino acid content, which might lead to an increase in total alkaloid content in the tubers and bulbils of *P*. *ternata*. 

Rudell et al. [37] reported that phytosterol metabolism was affected by ethylene in apple peel. BR treatment promotes the accumulation of ethylene in plants [38], which might affect the β-sitosterol content in the tubers and bulbils of *P*. *ternata*. The increased total flavone was observed in this paper and other papers. Ahammed et al. [39] reported that 24-epibrassinolide treatment (0.05 mg/L) enhanced the total flavone content in tomatoes on the 21st day after treatments. Ahanger et al. [35] study showed that epibrassinolide treatments (0.48 mg/L) increased the flavonoid content in tomato seedlings by 46.15% compared with the control. In the present study, total flavone content in the tubers and bulbils of *P*. *ternata* was increased by BR treatments. Ghassemi-Golezani et al. [40] found that 24-epibrassinolide increased the secondary metabolites of rape. The biosynthesis and signal transduction mechanisms of BR were closely related to the biosynthesis and signal transduction pathways of other plant hormones [9]. These complex regulatory mechanisms regulated plant growth and metabolism, which affected the content of active ingredients in the tubers and bulbils of *P*. *ternata*.

PAL and GS are important enzymes in plants, which play a key role in the phenylpropanoid pathway and process of inorganic N converted to organic, respectively. PAL catalyzes the deamination of amino groups from L-phenylalanine to produce *trans*-cinnamic acid, which is the substrate of alkaloid synthesis in *P*. *ternata* [41,42]. We found that PAL activity in the tubers and bulbils of *P*. *ternata* was significantly increased by BR treatments. This is in line with our results that BR treatments significantly increased alkaloid content in the tubers and bulbils of *P*. *ternata*. Ahammed et al. [13] reported that 24-epibrassinolide treatment (0.50 mg/L) had slight effects on the PAL activity in *Cucumis sativus* L. on the 10th day after treatments. The reason for this might be that the content of secondary metabolites in medicinal plants is higher than that in vegetables, so the effect of BR treatment on PAL activity in *P*. *ternata* were greater than those in *Cucumis sativus* L. GS catalyzes the formation of glutamine from glutamate in plants and plays an important role in the synthesis of free amino acids. In the current study, GS activity in the bulbils of *P*. *ternata* was increased in the early stage (15th, 45th, and 60th day) after BR treatments, and was decreased in the later stage (75th, 90th, and 105th day) after BR treatments. The reason for this might be that the tubers and bulbils of *P*. *ternate* needed to synthesize a large amount of the free amino acid content in the early growth stage, and the free amino acids were synthesized into total alkaloids in the late growth stage.

## 4. Materials and Methods

### 4.1. Plant Material and Experimental Design

The experiment was carried out in Hebei University, Baoding, China. The seed bulbs of *P*. *ternata* were obtained from Tianshui Chinese herbal medicine planting base, Gansu province, China. Seed bulbs of same size (diameter: 0.5–1.0 cm) were selected and planted into the boxes. Six rows were planted in each box (300 × 100 cm), with an average spacing of 18 cm. Each box was planted with about 900 seed bulbs. Routine management was conducted during the growth of *P*. *ternata*. 

### 4.2. Brassinolide Treatment

BR was initially dissolved in ethanol and made to volume with distilled water containing 0.02% Tween 20 (*v*/*v*) as an adhesive agent. *P*. *ternata* seed bulbs were cultivated until the three leaves fully expanded. Then six levels of BR (0.00, 0.05, 0.10, 0.50, 1.00, and 2.00 mg/L) were applied to them. All the treatments were applied using a manual sprayer in the afternoon for two consecutive days to strengthen their effects. Each treatment had four blocks, each of which was sprayed with 900 mL of solutions. Each plant was sprayed with about 1 mL of BR solution. The samples were collected on the 15th, 45th, 60th, 75th, 90th, and 105th days after BR treatments.

### 4.3. Tuber Yield

The tubers of P. ternata were randomly selected from each treatment and the soil was rinsed off before they were weighed.4.4. Total Flavone, Total Alkaloid and β-Sitosterol.

Total flavone content was determined by the method of Shi et al. [43] with some modifications. Dried samples (0.2 g) were extracted with 50% ethanol (5 mL) solution in an ultrasonic bath for 30 min. The reaction included 2.6 mL extract solution, 0.8 mL acetic acid-sodium acetate (pH 5.5), and 1.6 mL AlCl_3_ solution (1.5%). After 30 min, the mixture was measured at 415 nm using a spectrophotometer. Total flavone content was expressed as mg rutin g^−1^ dried weight (DW).

Total alkaloid content was measured according to the method described by Yu et al. [44]. The 0.2 g dried sample was added to 0.5 mL of 25% ammonium hydroxide and 5 mL chloroform. The mixtures were allowed to stand in a water bath (37 °C) for one hour. The reaction mixture contained 200 μL extract, 5 mL of citric acid-sodium citrate (pH 5.4), 0.5 mL of 0.1% bromothymol blue, 4.8 mL chloroform, and was kept at room temperature for one hour. Total alkaloid content was estimated by measuring at 416 nm and was expressed as mg g^−1^ DW.

β-sitosterol content was determined according to the method of Zhang et al. [45]. The 0.2 g dried sample was extracted with 10 mL of ethyl acetate. The mixture was allowed to stand for 12 h, and was centrifuged at 3400× *g* for 5 min. The extract (5 mL) was added to a 10 mL flask, and then the solution was evaporated in a water bath. The sediment was dissolved by sulfuric acid (5 mL), and then the solution was incubated at 50 °C for 4 min. The resulting complex solution was then measured at 407 nm, and results were expressed as mg g^−1^ DW.

### 4.4. Soluble Protein, Free Amino Acids and Soluble Sugar

Soluble protein content was determined according to the Bradford method [46]. Free amino acid content was measured according to the ninhydrin method [47]. Soluble sugar content was measured using a method with anthrone colorimetry [48]. 

### 4.5. Ascorbic Acid and DPPH Radical Scavenging

Ascorbic acid content was determined according to the method described by Yao et al. [49]. The dried sample (0.2 g) was extracted with 8 mL of oxalic acid (1%) in the ultrasonic bath for 30 min, followed by centrifugation at 9000× *g* for 5 min. Active carbon (0.2 g) was added to 5 mL supernatant and was filtered. The reaction solution included 1 mL supernatant, 1 mL of 2% thiourea, and 0.5 mL 2, 4-dinitrophenylhydrazine solution, and was kept at a temperature of 37 °C for three hours in a thermostatic bath. After cooling in an ice bath, 2.5 mL of 85% sulfuric acid was gradually added to the reaction solution. The absorbance value was measured at 500 nm, and the content was expressed as mg g^−1^ DW.

The 2,2-diphenyl-1-picrylhydrazyl (DPPH) radical activity was performed based on a procedure from the literature with a few modifications [50]. The sample extracted (2 mL) was added to 2 mL DPPH (0.04 g/L) in absolute ethanol. The mixture was left in the dark for 30 min and the absorbance was then measured at 517 nm, which was marked as A1. The control was made by adding 2 mL absolute ethanol to DPPH instead of the sample and was measured as A_0_. As for A_2_, the DPPH solution in A1 was replaced by absolute ethanol. DPPH radical scavenging activity (%) = [A_0_ − (A_1_ − A_2_/2)]/A_0_ × 100.

### 4.6. Phenylalanine Ammonia-Lyase (PAL) Activity

PAL activity was determined according to Shi et al. [43]. The fresh sample (0.3 g) was extracted with 4 mL of boric acid buffer (0.05 M, pH 8.7) containing 1% polyvinyl pyrrolidone and 5 mM β-mercaptoethanol, and was centrifuged at 7000× *g* for 15 min at 4 °C. The reaction solution included 0.5 mL of enzyme extract and 3 mL of phenylalanine (0.02 M), and was incubated at 30 °C for 30 min. The reaction was stopped by adding 0.5 mL of HCl (6 M). The absorbance was measured at 290 nm, and PAL activity was expressed as A g^−1^ h^−1^ fresh weigh (FW).

### 4.7. Glutamine Synthetase (GS) Activity

GS activity was determined according to the method described by Yao and Liu [51]. The fresh sample (0.3 g) was extracted with 3 mL of 0.05 M Tris-HCL (PH 8.0), containing 2 mM Mg^2+^, 0.4 M sucrose and 2 mM dithiothreitol. Extracts were centrifuged at 7000× *g* for 15 min at 4 °C. The reaction solution included 0.7 mL of supernatant, 0.7 mL of ATP (40 mM), and 1.6 mL of hydroxylamine amine hydrochloride (80 mM, pH 7.4), and was incubated at 37 °C for 30 min. The reaction was terminated by the addition of 1 mL of 0.37 M FeCl_3_ solution and was centrifuged at 3400× *g* for 5 min. The absorbance of the solution was read at 540 nm. The GS activity was expressed as A540 g^−1^ h^−1^ FW.

### 4.8. Statistical Analysis

Four replicates per treatment were carried out and all statistical analyses were performed using the software Statistical Package for Social Sciences (SPSS) version 26.0 (IBM, Chicago, IL, USA). The effect of BR on tubers and bulbils was analyzed using variance (ANOVA), followed by Tukey’s test at a significance level of *p* < 0.05. All results are expressed as the mean ± standard error. The principal component analysis (PCA) was used to calculate scores for the tubers and bulbils of *P*. *ternata* during growth stages after BR treatments. The initial eigenvalue of >1 was used as the basis for determining the number of principal components. 

## 5. Conclusions

We found that application of BR significantly affected the tuber yield, active ingredients, and nutrient components in the tubers and bulbils of *P*. *ternata*, and the response to BR concentration was different at different growth stages. The tuber yield of *P*. *ternata* was increased by BR treatments at different stages (except for the 15th day after treatments). On the 75th day after BR treatments, optimal BR concentration not only significantly improved the yield of *P*. *ternata*, but also meant that *P. ternata* retained high levels of total alkaloids in tubers and bulbils, high total flavone content in tubers and bulbils, and high β-sitosterol content in tubers of *P*. *ternata*. We chose the optimal model for harvesting time after different concentrations of BR treatment, which provides a way to improve the quality and yield of medicinal plants at the same time, although further research is required to reveal its mechanism.

## Figures and Tables

**Figure 1 ijms-23-15375-f001:**
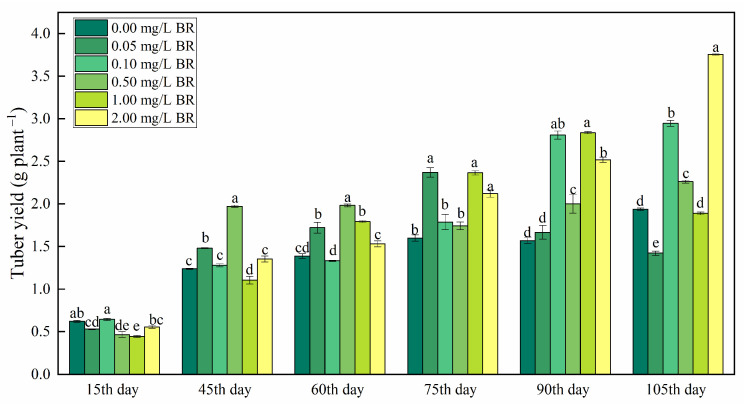
The change in tuber yield of *P. ternata* at different stages after BR treatments. The bars with different letters are significantly different from each other at the same stage after BR treatments (*p* < 0.05). Values are means of four replicates ± SE.

**Figure 2 ijms-23-15375-f002:**
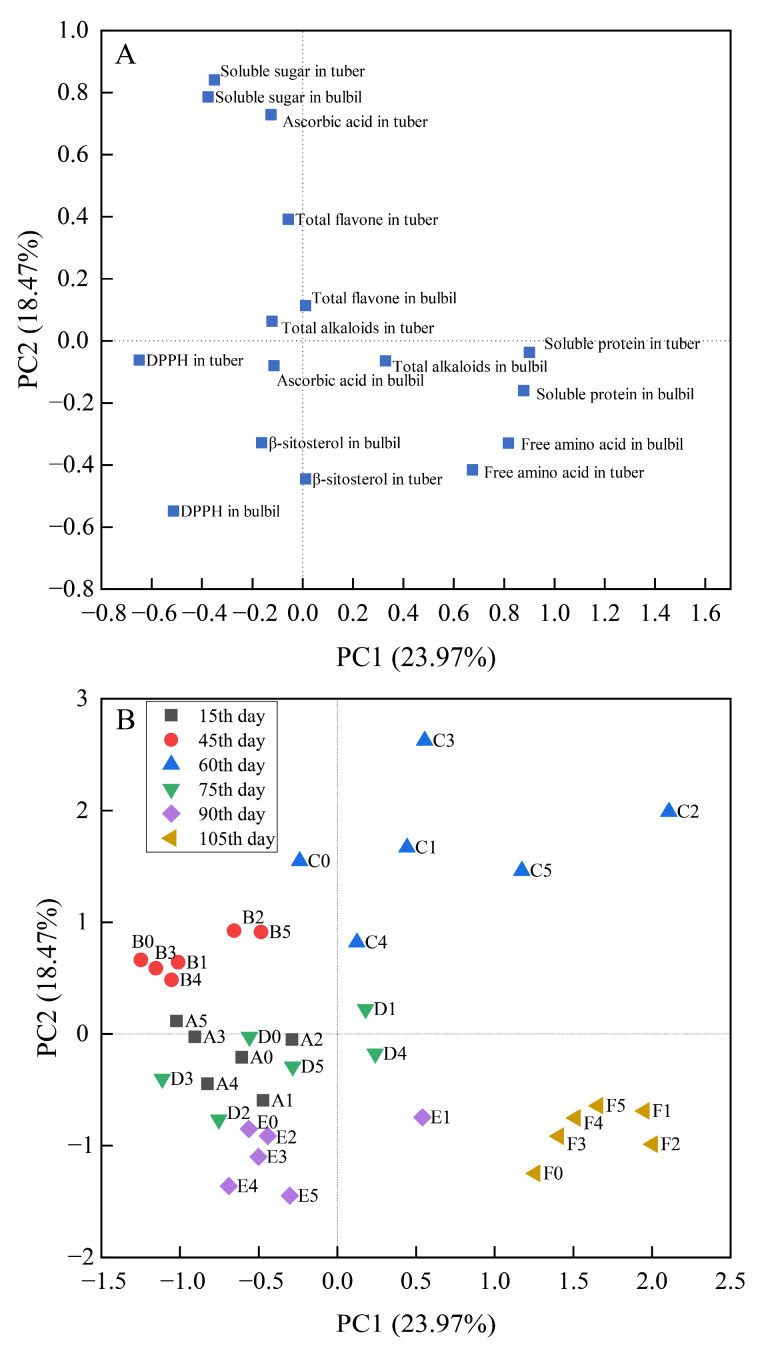
Principal component analysis (PCA) loading (**A**) and scores (**B**) showing correlation of yield and quality parameters in tubers and bulbils of *P. ternata* with PC1-PC2 at different stages after BR treatments. A, 15th day; B, 45th day; C, 60th day; D, 75th day; E, 90th day; F, 105th day. 0, 0.00 mg/L BR; 1, 0.05 mg/L BR; 2, 0.10 mg/L BR; 3, 0.50 mg/L BR; 4, 1.00 mg/L BR; 5, 2.00 mg/L BR. PCA was performed on the correlation matrix of values of yield and quality in the tubers and bulbils of *P. ternata*.

**Figure 3 ijms-23-15375-f003:**
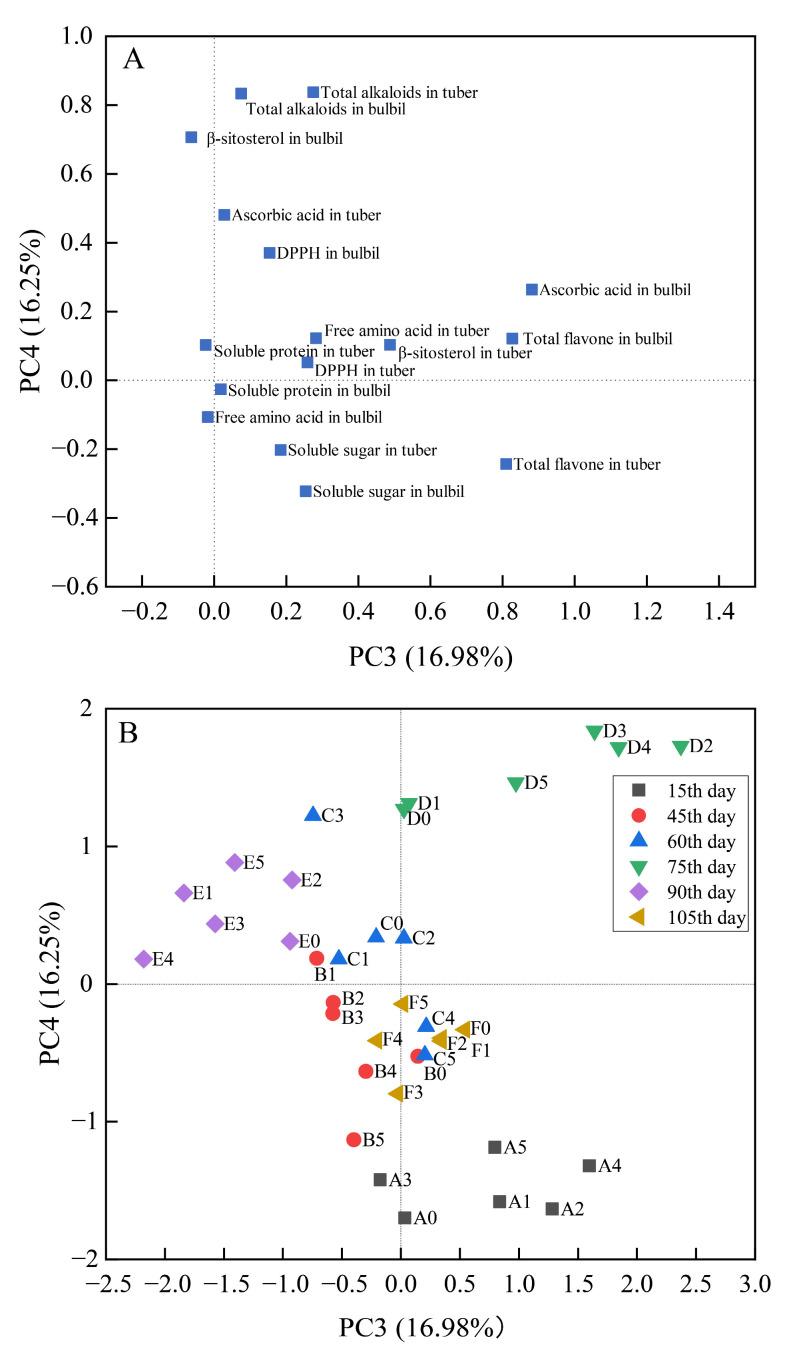
Principal component analysis (PCA) loading (**A**) and scores (**B**) showing correlation of yield and quality parameters in tubers and bulbils of *P. ternata* with PC3-PC4 at different stages after BR treatments. A, 15th day; B, 45th day; C, 60th day; D, 75th day; E, 90th day; F, 105th day. 0, 0.00 mg/L BR; 1, 0.05 mg/L BR; 2, 0.10 mg/L BR; 3, 0.50 mg/L BR; 4, 1.00 mg/L BR; 5, 2.00 mg/L BR. PCA was performed on the correlation matrix of values of yield and quality in the tubers and bulbils of *P. ternata*.

**Figure 4 ijms-23-15375-f004:**
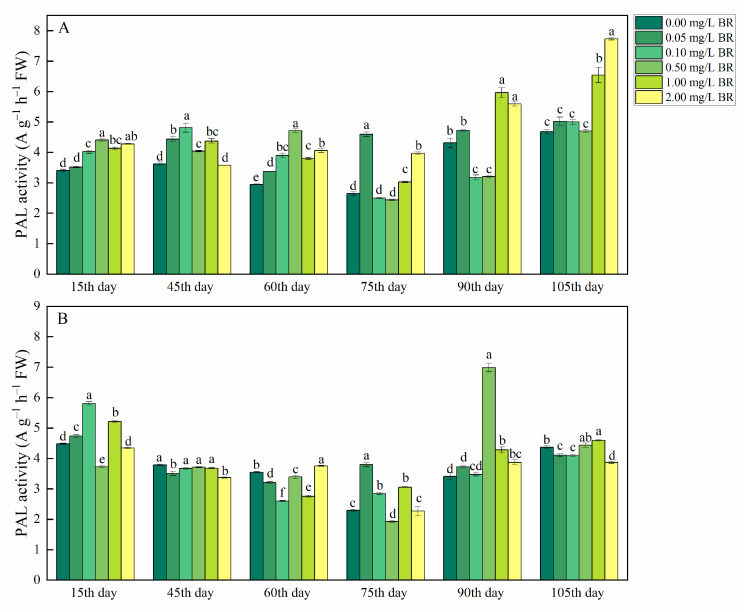
The change in PAL activity in tubers (**A**) and bulbils (**B**) of *P. ternata* at different stages after BR treatments. The bars with different letters are significantly different from each other at the same stage after BR treatments (*p* < 0.05). Values are means of four replicates ± SE.

**Figure 5 ijms-23-15375-f005:**
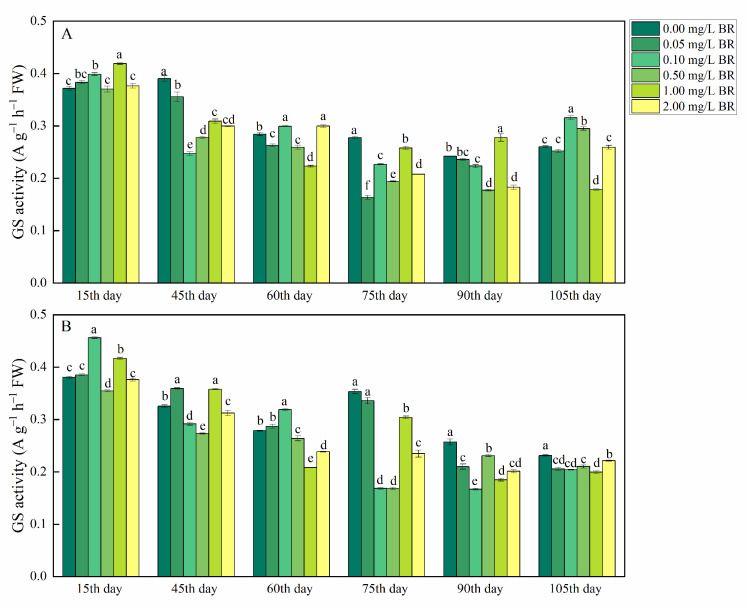
The change in GS activity in tubers (**A**) and bulbils (**B**) of *P. ternata* at different stages after BR treatments. The bars with different letters are significantly different from each other at the same stage after BR treatments (*p* < 0.05). Values are means of four replicates ± SE.

## Data Availability

Not applicable.

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
