# Peer review of "Principal Component Analysis to Assess the Changes of Yield and Quality in Pinellia ternata at Different Stages after Brassinolide Treatments"

_ijms, 2022, doi:10.3390/ijms232315375_

Round 1

Reviewer 1 Report

Manuscript "Principal component analysis to assess the changes of yield and quality in Pinellia ternata during different stages after brassinolide treatments" by authors:

Chenchen Guo, Yanfen Zhang, Dengyun Wu, Mengyue Wang, Yu Du, Jianzhou Chu, Xiaoqin Yao is a manuscript on the role of brasinosteroids in the ontogeny of the important medical plant triad. The manuscript is well structured, contains new interesting information. However, it needs some significant fixes.

So in the manuscript there is no conclusion with a consideration of the prospects and an indication of the priority.

In figure 2a, the inscriptions go beyond the edge of the image.

The appearance of the histograms would be better increased.

Author Response

Manuscript "Principal component analysis to assess the changes of yield and quality in Pinellia ternata during different stages after brassinolide treatments" by authors:

Chenchen Guo, Yanfen Zhang, Dengyun Wu, Mengyue Wang, Yu Du, Jianzhou Chu, Xiaoqin Yao is a manuscript on the role of brasinosteroids in the ontogeny of the important medical plant triad. The manuscript is well structured, contains new interesting information. However, it needs some significant fixes.

  1. So in the manuscript there is no conclusion with a consideration of the prospects and an indication of the priority.

Yes, we added the conclusion in revised paper. “We chose the optimal model for harvesting time after different concentrations of BR treatment, which provides a way how to improve the quality and yield of medicinal plants at the same time, although further research is required to reveal its mechanism.”

  1. In figure 2a, the inscriptions go beyond the edge of the image.

Yes, we changed the figure 2a in revised paper.

  1. The appearance of the histograms would be better increased.

Yes, we changed the histograms in revised paper.

Reviewer 2 Report

The article considers the effect of treatment of the medicinal plant Pinellia ternata with several concentrations of brassinolide (BR) on tuber and bulbil yield and their biochemical components. It is shown that at different stages after treatment with brassinolide (from 15 to 105 days), the effects of treatment were manifested significantly, increasing the yield and affecting the content of major primary and secondary metabolites. The article contains some new information, but there are several questions and comments. 

1. There is a lot of information on P. ternata reproduction, yield, and chemical components in Introductions. However, there is no clear articulation of the major problems associated with these processes in P. ternata.  Also missing a rationale for selecting BRs for plant treatments.  

2. Phrases on lines 46-47, 56-58, 60-61 contain axioms absolutely applicable to all plants that do not require clarification or proof for P. ternata. In general, it is unclear what the reasoning on lines 46-50 is for. 

3. Section 4. Materials and methods does not contain the necessary information. How many plants were grown in one box? What was the age of the plants when they were treated with BR? How many mL of BR solution was used to spray 1 plant? 

4. It is not clear why the tuber yield (Fig. 1) varied sharply at different periods of time with different doses of BP, often giving a negative increase in dynamics. Obviously, this is due to errors in tuber sampling for dry weight analysis. 

5. The PCA assay (Figs. 2, 3) is described quite curtly. 

6. The authors do not explain why, of the many enzymes involved in the synthesis of alkaloids, flavones, sitosterol and other low molecular weight compounds analyzed in the article, only PAL and HS activity changes were analyzed. A rationale needs to be given in the Introduction. 

7. Line 248 contains an incorrect judgment: "Total alkaloids are mainly composed of amino acids". The initial steps of alkaloid synthesis come from amino acids, but the alkaloids themselves are not composed of amino acids. 

8. The flow of information should be given more clearly throughout the article. 

Author Response

The article considers the effect of treatment of the medicinal plant Pinellia ternata with several concentrations of brassinolide (BR) on tuber and bulbil yield and their biochemical components. It is shown that at different stages after treatment with brassinolide (from 15 to 105 days), the effects of treatment were manifested significantly, increasing the yield and affecting the content of major primary and secondary metabolites. The article contains some new information, but there are several questions and comments.

  1. There is a lot of information on P. ternata reproduction, yield, and chemical components in Introductions. However, there is no clear articulation of the major problems associated with these processes in P. ternata. Also missing a rationale for selecting BRs for plant treatments.

Yes, we added the major problems associated with P. ternata study and a rationale for selecting BRs for plant treatments. “Driven by the wide functions of P. ternata, the huge demand makes it in short supply. However, there are few studies on improving the yield and quality of P. ternata. For example, light intensities and temperatures impacted the growth and tube yield of P. ternata [7, 8]. P. ternata is mainly asexual reproduction. Besides, there are few studies on the bulbil of P. ternata, which is a crucial propagative material of P. ternata. The wild resource of P. ternata is rare and the technology is also deficient in artificial cultivation of P. ternata. Therefore, it is necessary to improve the yield and quality of P. ternata at the same time.” “P. ternata is susceptible to withered caused by environmental stress, resulting in the decline of yield and quality of P. ternata.” “Several studies have shown that BR increased plant growth under various environmental stresses,” “Understanding the effect of different BR concentrations on the yield and quality of P. ternata is very important in field cultivation.”

  1. Phrases on lines 46-47, 56-58, 60-61 contain axioms absolutely applicable to all plants that do not require clarification or proof for P. ternata. In general, it is unclear what the reasoning on lines 46-50 is for.

Yes, we rewrote it in revised paper.

  1. Section 4. Materials and methods does not contain the necessary information. How many plants were grown in one box? What was the age of the plants when they were treated with BR? How many mL of BR solution was used to spray 1 plant?

  Yes, we added more information in Section 4. Materials and methods. “Each box was planted about 900 seed bulbs.” “P. ternata seed bulbs were cultivated until the three-leaf fully expanded. Then it was applied with six BR levels (0.00, 0.05, 0.10, 0.50, 1.00, and 2.00 mg/L).” “Each plant was sprayed with about 1mL of BR solution.”

  1. It is not clear why the tuber yield (Fig. 1) varied sharply at different periods of time with different doses of BP, often giving a negative increase in dynamics. Obviously, this is due to errors in tuber sampling for dry weight analysis.

Yes, we measured the fresh weight of the P. ternata tubers. After the BR treatment, the tubers yield of P. ternata tended to increase with the growth time. In addition, after the 15th, 45th, 60th, 75th, 90th, and 105th of BR treatments, the tuber yield of P. ternata reached the peak value at 0.10, 0.50, 0.50, 1.00, 1.00, and 2.00 mg/L BR, respectively. The reason may be that high BR treatment can be maintained for a longer stage of time to promote P. ternata tuber yield.

  1. The PCA assay (Figs. 2, 3) is described quite curtly.

Yes, we described the loadings and scores of the yield and quality parameters in tuber and bulbil of principal components 1, 2, 3, and 4. The results of the PCA were discussed in detail in the discussion section.

  1. The authors do not explain why, of the many enzymes involved in the synthesis of alkaloids, flavones, sitosterol and other low molecular weight compounds analyzed in the article, only PAL and HS activity changes were analyzed. A rationale needs to be given in the Introduction.

Yes, we added the rationale for measuring these two enzymes in the discussion. “PAL and GS are important enzymes in plants, which play a key role in the phenylpropanoid pathway and process of inorganic N converted to organic, respectively. PAL catalyzes the deamination of amino groups from L-phenylalanine to produce trans-cinnamic acid, and that is the substrate of alkaloid synthesis in P. ternata [41, 42].” “GS catalyzes the formation of glutamine from glutamate in plants and plays an important role in the synthesis of free amino acids.” “Total alkaloids are mainly synthesis come from amino acids, and their biosynthesis includes multistep reactions [36].”

  1. Line 248 contains an incorrect judgment: "Total alkaloids are mainly composed of amino acids". The initial steps of alkaloid synthesis come from amino acids, but the alkaloids themselves are not composed of amino acids.

Yes, we revised it in revised paper. “The initial steps of alkaloid synthesis come from amino acids, and their biosynthesis includes multistep reactions [36]”

  1. The flow of information should be given more clearly throughout the article.

Yes

Reviewer 3 Report

The article Principal component analysis to assess the changes of yield and quality in Pinellia ternata during different stages after brassinolide treatments by Chenchen Guo et al examines the effective use of epibrasinolide to improve the quality and pyroproductivity of Pinellia ternata. The use of growth regulators is extremely important for crop stability and a detailed study of practical application is relevant.

The manuscript is carefully prepared, but there are a number of comments that need to be corrected before submission. For example, at the end of the introduction, the authors write that they put forward a hypothesis, but they probably meant that in the article they are testing the hypothesis. Whether this is a hypothesis, and not just a test of the assumption with the choice of the optimal mode, is also doubtful. This passage should be rephrased.

The introduction does not reflect the safety aspect of the use of steroids for consumers and staff, which is not obvious.

The hypothesis about the method of influencing quality is also little understood. After all, the composition of the fruit after processing was not checked.

The design of the article also leaves much to be desired, black-and-white columns in a full-color magazine look like an anachronism.

Figure 2 letters are poorly readable and signs are in some cases untidy located.

It is not clear why there are no images of treated and untreated plants and their products, tubers, leaves.

The discussion does not contain a section on the safety of the obtained raw materials, because significant changes in metabolism can lead to a significant change in medicinal properties. It is not clear how it is proposed to take into account its potential benefit or danger.

There is no conclusion allocated in a separate section in the article.

The article needs revision and correction.

Author Response

The article Principal component analysis to assess the changes of yield and quality in Pinellia ternata during different stages after brassinolide treatments by Chenchen Guo et al examines the effective use of epibrasinolide to improve the quality and pyroproductivity of Pinellia ternata. The use of growth regulators is extremely important for crop stability and a detailed study of practical application is relevant.

  1. The manuscript is carefully prepared, but there are a number of comments that need to be corrected before submission. For example, at the end of the introduction, the authors write that they put forward a hypothesis, but they probably meant that in the article they are testing the hypothesis. Whether this is a hypothesis, and not just a test of the assumption with the choice of the optimal mode, is also doubtful. This passage should be rephrased.

Yes, we changed the hypothesis in revised paper. “We chose the optimal model for harvesting time after different concentrations of BR treatment, which would better meet the artificial cultivation of P. ternata on improving yield.”

  1. The introduction does not reflect the safety aspect of the use of steroids for consumers and staff, which is not obvious.

Yes, we added the safety aspect of BR in revised paper. “BRs can inhibit the growth of several human cancer cell lines (breast and prostate) at micromolar concentrations without affecting the growth of normal cells [18]. BR is non-toxic and prevents the growth of cancer cells, they can be used in field cultivation without causing environmental pollution.”

  1. The hypothesis about the method of influencing quality is also little understood. After all, the composition of the fruit after processing was not checked.

Sorry, we did not measure the composition of the fruit after processing. P. ternata is a medicinal plant and the quality is mainly the medicinally active ingredient. We investigated the effect of BR treatments on tuber yield and medicinal active ingredients (alkaloids, β-Sitosterol, and flavonoids) during the growth of P. ternata. Thanks again.

  1. The design of the article also leaves much to be desired, black-and-white columns in a full-color magazine look like an anachronism.

Yes, we changed it in revised paper.

  1. Figure 2 letters are poorly readable and signs are in some cases untidy located.

Yes, we changed Figure 2 in revised paper.

  1. It is not clear why there are no images of treated and untreated plants and their products, tubers, leaves.

Sorry, there was no significant difference in the appearance of the P. ternata tubers and leaves between BR treated and untreated. So, we did not put images of the appearance in the paper. Thanks again!

  1. The discussion does not contain a section on the safety of the obtained raw materials, because significant changes in metabolism can lead to a significant change in medicinal properties. It is not clear how it is proposed to take into account its potential benefit or danger.

Yes, the tuber and bulbil of P. ternata contain alkaloids, β-Sitosterol, flavonoids, lectins, volatile oils, fatty acids, etc., among which alkaloids are most important active components. BR treatments increased the content of alkaloids, β-Sitosterol, and flavonoids and thus improved the quality of raw materials.

  1. There is no conclusion allocated in a separate section in the article.

Yes, we added the conclusion in revised paper.

  1. The article needs revision and correction.

Yes, we revised and corrected this article. Thanks again!

Round 2

Reviewer 2 Report

The corrected manuscript contains all the necessary information. The authors have fully taken into account all the comments and have made the necessary changes in the text of the article. The article can be accepted for publication.

Reviewer 3 Report

The article Principal component analysis to assess the changes of yield and quality in Pinellia ternata during different stages after brassinolide treatments by Chenchen Guo et al examines the effective use of epibrasinolide to improve the quality and pyroproductivity of Pinellia ternata . 

After the changes made, the concept of the authors became more understandable and relevant.

The manuscript may be published in its present form.